# Multi-Source Multi-Type Knowledge Exploration and Exploitation for Dialogue Generation

**Xuanfan Ni[1,2], Hongliang Dai[1,2], Zhaochun Ren[3], Piji Li[1,2*]**
[1]College of Computer Science and Technology,
Nanjing University of Aeronautics and Astronautics, China
[2]MIIT Key Laboratory of Pattern Analysis and Machine Intelligence, Nanjing, China
[3]Shandong University, Qingdao, China
[1]{xuanfanni,hongldai,pjli}@nuaa.edu.cn
[3]zhaochun.ren@sdu.edu.cn

## Abstract

Open-domain multi-turn dialogue generation encounters the significant challenge of lacking various types of knowledge from diverse sources. Existing models typically focus on identifying specific types of dialogue knowledge and utilize corresponding datasets for training. However, this approach often leads to limited generalization capabilities and increased computational resource requirements. Recently, large language models (LLMs) have shown impressive performance on natural language processing tasks. To harness the knowledge storage of LLMs, we propose a framework named KnowEE that explores multi-source multi-type knowledge from LLMs by leveraging diverse datasets and then exploits the obtained knowledge for response generation. Our framework comprises two phases: First, we leverage five external datasets encompassing various types of knowledge to extract the most relevant samples to the dialogue context which are served as prompts to generate corresponding type of knowledge; Second, we inject the acquired knowledge into the ongoing dialogue context in fine-grained and coarse-grained manners, which is then fed into LLMs to generate the final dialogue response. Both automatic and manual evaluation results validate the effectiveness of our framework in exploring and exploiting multi-source multi-type knowledge to generate coherent, informative, and fluent responses.[1]

## 1 Introduction

Open-domain multi-turn dialogue generation necessitates models capable of producing high-quality responses that are coherent, consistent, and informative (Li et al., 2016b; Roller et al., 2021; Wu et al., 2022). In human conversations, we naturally connect various **types** of knowledge such **emotion**,

---

*Corresponding author.
[1]Code: https://github.com/Patrick-Ni/KnowEE

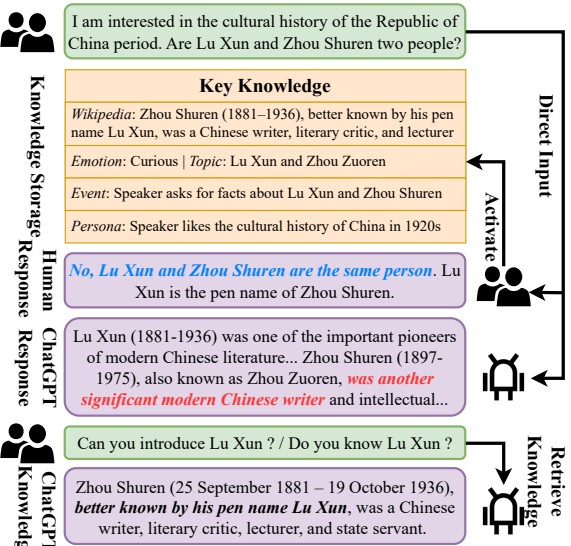

Figure 1: Two cases of dialogue generation for human and ChatGPT. When generating responses, humans employ their knowledge related to the dialogue history to ensure coherency and accuracy. Conversely, solely inputting the dialogue history into ChatGPT can lead to hallucination, despite its possession of corresponding knowledge.

**topic**, **event**, **persona**, **general world knowledge**, etc. to facilitate smooth and high-quality communication. And this knowledge is gradually acquired through various **sources** in the process of learning after birth. For example, as depicted in Figure 1, when asked if *Lu Xun* and *Zhou Shuren* are the same person, humans draw upon various knowledge sources to comprehend different types of knowledge related to the dialogue history, including the speaker's emotion and intent, the overall topic of the dialogue, and relevant background knowledge about *Lu Xun* and *Zhou Shuren*. Finally, we can provide a appropriate and correct response. Therefore, enabling a model to explore and aggregate various types of knowledge from multiple sources, similar to how humans do, and exploiting them effectively will play a crucial role in im-

proving the quality of intelligent human-machine conversations (Wu et al., 2022).

Previous studies have utilized external knowledge sources to bridge the knowledge gap between machines and humans in conversation by injecting knowledge that is difficult to learn purely from the given dialogue training dataset (Ghazvininejad et al., 2018; Wu et al., 2021; Xu et al., 2022). However, these existing knowledge-enhanced dialogue generation approaches have several notable problems: (1) limited knowledge types, (2) monotonous knowledge sources, and (3) complicated and inefficient knowledge mining and fusion strategies. For example, incorporating external knowledge of specific type into the training process is a common strategy, such as commonsense and emotion (Li et al., 2022), topic (Xu et al., 2021), persona (Yin et al., 2023), Wikipedia knowledge (Zhao et al., 2020), etc. Nevertheless, these approach often suffers from poor generalization due to constraints in the knowledge types and sources. Furthermore, the process of modeling knowledge for training or inference introduces additional computational resource overhead. Another approach is to use search engines to generate responses based on internet search results (Komeili et al., 2022). While this reduces training and inference costs, it fails to enable the model to fully understand the dialogue history as the knowledge is not stored internally.

In addition, several researchers have endeavored to utilize large language models (LLMs) for dialogue generation tasks. These models, such as GPT-4 (OpenAI, 2023), have exhibited exceptional performance across various natural language processing tasks (Yasunaga et al., 2021; Zhang et al., 2023; Ni et al., 2023), and show two significant capabilities to (1) store and access various types of knowledge from extensive training sources (Zhao et al., 2023), and (2) perform different downstream tasks without fine-tuning. Intuitively, the knowledge storage and inference capacity as well as the text understanding and generation capabilities of LLMs can help generate better conversations, even for the zero-shot or few-shot scenarios. Nevertheless, despite these superior capabilities, even the most advanced LLMs such as ChatGPT, often encounter challenges in dialogue generation tasks, as depicted in Figure 1. Although ChatGPT "knows" that *Lu Xun* is a pseudonym for *Zhou Shuren*, it fails to accurately distinguish between the two persons when directly input the dialogue context into the ChatGPT system. We analyze that relying solely on the dialogue historical context cannot effectively stimulate and explore the rich knowledge stored within the LLMs when generating responses.

To maximize the utilization of the rich knowledge stored in LLMs for dialogue generation system, we propose a framework named **KnowEE**, which can conduct multi-source multi-type **Know**ledge **E**xploration from LLMs and then conduct knowledge **E**xploitation for better dialogue understanding and response generation. As mentioned above, in our work we identify the dialogue knowledge into five types: *Emotion*, *Topic*, *Persona*, *Event*, and *General World knowledge*. Our framework KnowEE includes two phases: *multi-source multi-type knowledge exploration via in-context learning* and *fine/coarse-grained knowledge exploitation for response generation*. For the first phase, given the dialogue context, we leverage five external source datasets with knowledge labels corresponding to the above five types of knowledge, to extract the most relevant samples to the given context. These samples will serve as prompts to explore the LLMs to generate knowledge for the corresponding type. Next, we inject the generated knowledge into ongoing dialogue context using two different approaches: utterance-level fine-grained knowledge injection (FgKI) and dialogue-level coarse-grained knowledge injection (CgKI). The combined information is then fed into LLMs to obtain the final response. We evaluated our proposed framework KnowEE on four dialogue generation tasks: EMPATHETICDIALOGUES (ED), DAILYDIALOG (DD), PERSONA-CHAT (PC) and OPENDIALKG (ODKG), and the experimental results demonstrate that it outperforms the baseline methods in terms of both automatic and manual evaluation metrics.

Our contributions can be summarized as follows:

- We propose a framework named KnowEE that explores multi-source multi-type knowledge from large language models using diverse source datasets, and exploits the knowledge in fine-grained and coarse-grained manners for dialogue generation.
- We leverage five extrenal datasets and extract the most relevant samples to the dialogue context, and prompt the LLMs to generate dialogue knowledge via in-context learning. Additionally, we propose a fine-grained and a coarse knowledge injection approaches to

combine the generated knowledge with the dialogue context.

- The automatic and manual evaluation on four datasets shows that our framework is superior to the strong zero-shot and few-shot baselines in terms of perplexity and diversity, and capable of generating more fluent, coherent, and informative responses.

## 2 Related Work

### 2.1 Open-domain Dialogue Generation

The task aims to establish long-term connections and provide communication satisfying human need (Ghazarian et al., 2019; Pan et al., 2019; Huang et al., 2020; Aliannejadi et al., 2021). Previous works often leverage neural network model and identify one type of dialogue, such as emotion (Li et al., 2020; Hu et al., 2021; Li et al., 2022), topic (Shi et al., 2016; Zhu et al., 2021; Xu et al., 2021) and persona (Liu et al., 2020; Yin et al., 2023), which have achieved impressive performance on corresponding datasets and arouse widespread interest. In addition to the widely used supervised learning, some researchers introduce other effective algorithms or means, like Reinforcement Learning (Saleh et al., 2020) and Contrastive Learning (Cai et al., 2020). Besides, the general LLMs, such as GPT-3 (Brown et al., 2020), GLM (Zeng et al., 2022) and OPT (Zhang et al., 2022), are found to provide competitive responses with more efficient solutions (Zheng and Huang, 2021). These observations motivate some researchers to explore the prompt engineering in LLMs for dialogue generation. Yu et al. (2022) proposes a knowledge-grounded dialogue system that is equipped with the prompt-aware tuning-free LLMs exploitation and supported by the ready-to-use open-domain external knowledge bases and search engines. In contrast, our work focuses on a wider variety of dialogue knowledge and leverages a richer external knowledge resources.

### 2.2 Large Language Models

Language models have significantly enhanced the performance of various NLP tasks owing to the benefits of pre-training and transformer-based structure (Jing and Xu, 2019; Qiu et al., 2020; Han et al., 2021; Li et al., 2021; Ni et al., 2023). With the groundwork of LLMs-based algorithms, researchers now focus on unfolding the capabilities of LLMs effectively and efficiently, which is a crucial challenge (Yu et al., 2022). A common and popular strategy for further training LLMs on downstream tasks is fine-tuning (Sun et al., 2022). However, with the rapid growth of model scale, LLMs require an abundance of high-quality corpus and expensive computational resources for a single fine-tuning, rendering them impractical to use. To address these challenges, novel techniques such as prompt learning (Liu et al., 2023) and prefix-tuning (Li and Liang, 2021) have been proposed. These techniques fine-tune only a few parameters instead of the whole, with the expectation of achieving comparable performance to fully fine-tuned models. Besides, in-context learning (Brown et al., 2020), which involves inputting exemplars related to downstream tasks to models rather than additional training, has proven to be effective in few-shot settings (Chan et al., 2022; Rubin et al., 2022).

## 3 Methodology

### 3.1 Overview

Figure 2 depicts the workflow of our framework KnowEE. The input comprises a set of conversational utterances, formally represented as $\mathcal{U}_t = \{U_1, S_1, \ldots, U_{t-1}, S_{t-1}, U_t\}$. Here, $U_i$ and $S_i$ denote the $i^{th}$ utterances from different speakers. The objective of the dialogue generation task is to generate a response $S_t$ for the $t^{th}$ round of the dialogue. Recognizing the fact that humans utilize various types of knowledge from diverse sources to formulate responses, we adopt a two-step generation process: *multi-source multi-type dialogue exploration* and *knowledge exploitation for response generation*. First, we employ external datasets to explore the LLMs to generate five types of knowledge relevant to the dialogue historical context: Emotion, Topic, Persona, Event, and General World Knowledge, denoted as $K_{emo}$, $K_{tpc}$, $K_{psn}$, $K_{evt}$, and $K_{wor}$, respectively. Subsequently, we inject the generated knowledge into ongoing dialogue context and feed it into LLMs to generate high-quality responses. We also design two approaches with varying levels of granularity to optimize knowledge injection while adhering to the constraint of input length.

### 3.2 Multi-Source Multi-Type Knowledge Exploration

Given the dialogue historical context $\mathcal{U}_t$, the target of knowledge exploration is to produce all types of knowledge required for the response gen-

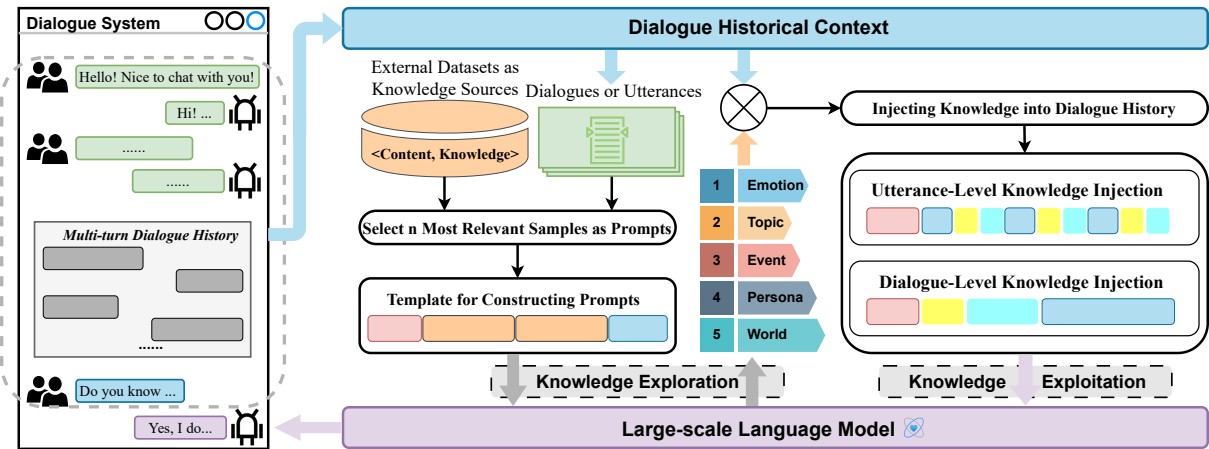

Figure 2: The overall architecture of our proposed framework, KnowEE. The direction of the arrows represent the process of dialogue response generation. The same content is marked with the same color.

eration procedure. We analyze numerous dialogue examples and identify the dialogue knowledge into five types: **Emotion**, **Topic**, **Persona**, **Event**, and **General World knowledge**. For each type, we select one labelled dataset correspondingly: GOEMOTIONS ($\mathbb{GE}$) (Demszky et al., 2020) for emotion information, DAILYDIALOG ($\mathbb{DD}$) (Li et al., 2017) for topic information, PERSONACHAT ($\mathbb{PC}$) (Zhang et al., 2018) for persona, ATOMIC ($\mathbb{AT}$) (Sap et al., 2019) for event, and WIZARD OF WIKIPEDIA ($\mathbb{WOW}$) (Dinan et al., 2019) for general world knowledge. Since these datasets gather a significant amount of human-labeled data, we can view them as knowledge sources $\{D_{emo}, D_{tpc}, D_{psn}, D_{evt}, D_{wor}\}$ containing a large number of content-knowledge pairs $\langle C, K \rangle$, where "content" refers to the textual data in the dataset, such as Reddit comments in $\mathbb{GE}$ or dialogue context in $\mathbb{DD}$; "knowledge" represents the specific type of knowledge corresponding to the "content", such as dialogue topics in $\mathbb{DD}$ or Wikipedia knowledge in $\mathbb{WOW}$. Table 1 gives the statistics of these datasets.

Then how to acquire those five types of knowledge based on the five source datasets for dialogue context $\mathcal{U}_t$? Considering the powerful knowledge storage capacity of large language models, we propose a prompt-based approach via in-context learning (Brown et al., 2020) to effectively stimulate and explore the rich knowledge stored within them. Our hypothesis posits that the selection of appropriate samples from source datasets as prompts is paramount in producing high-quality knowledge labels. Intuitively, incorporating knowledge from similar dialogue contexts can assist models in generating relatively accurate knowledge.

Hence, we employ a query-based sample selection method, wherein we utilize either the entire dialogue context or a single utterance as a query $Q$, to sample content-knowledge pairs $\langle C, K \rangle$ from all the source datasets, based on the desired level of knowledge granularity. Consequently, we obtain $\{Q_{emo}, Q_{tpc}, Q_{psn}, Q_{evt}, Q_{wor}\}$ corresponding to the above five knowledge sources. To ensure the relevance of the selected examples to the query, we utilize SentenceTransformer (ST) (Reimers and Gurevych, 2019) to encode the query and the pairs $\{\langle C_1^i, K_1^i \rangle, \langle C_2^i, K_2^i \rangle, \ldots, \langle C_m^i, K_m^i \rangle\}$ in $D_i$ ($i \in [emo, tpc, psn, evt, wor]$). Subsequently, we calculate the cosine similarity between the query $Q_i$ and $j^{th}$ pair ($j \in [1, m]$):

$$S_{sim}^{i,j} = \text{cosine}(ST(Q_i), ST(C_j^i))$$

For each source dataset, we select $n$ samples with the highest similarity scores to the query and use them to create in-context prompts. Specifically, the prompt $\mathcal{P}_r^i$ for the $r^{th}$ sample ($r \in [1, n]$) in $D_i$ is "$[C_r^i; T_i \Rightarrow K_r^i]$", and the prompt $\mathcal{P}_{query}^i$ for the query $Q_i$ is "$[Q_i; T_i \Rightarrow]$"[1], where $[;]$ denotes concatenation and $T_i$ represents the name of one of the five knowledge types, such as "emotion", "topic", etc., depends on the corresponding dataset $D_i$. In order to indicate the current task, we also include an instruction $\mathcal{I}_i$, with the template "*Please try to understand the input dialogue history and generate corresponding $T_i$ knowledge.*"

Finally, we concatenate the instruction and constructed prompts, and feed them into LLMs to generate the dialogue knowledge $K_i$:

$$K_i = \mathcal{LLM}([\mathcal{I}_i; \mathcal{P}_1^i; \mathcal{P}_2^i \ldots \mathcal{P}_n^i; \mathcal{P}_{query}^i])$$

---

[1]For example: I wanted to downvote this, but it's not your fault.  Emotion ⇒ Disappointment

| Type | Dataset | Num | Avg. CT | Avg. KT | Gran. |
|---|---|---|---|---|---|
| Emotion | $\mathbb{GE}$ | 211K | 20.1 | 1.0 | Fine |
| Topic | $\mathbb{DD}$ | 10K | 144.2 | 1.4 | Coarse |
| Persona | $\mathbb{PC}$ | 17K | 117.9 | 43.2 | Coarse |
| Event | $\mathbb{AT}$ | 24K | 8.9 | 5.4 | Fine |
| World Klg. | $\mathbb{WOW}$ | 166K | 22.9 | 35.9 | Fine |

Table 1: Statistics of datasets corresponding to five types of knowledge. Each content-knowledge pair is viewed as one data sample. **Avg. CT** and **Avg. KT** represent average content tokens and average knowledge tokens. **Gran** represents the granularity of textual data in the dataset and query from dialogue history.

Thus, we can obtain $K_{emo}$, $K_{tpc}$, $K_{psn}$, $K_{evt}$, and $K_{wor}$, which is related to dialogue historical context.

## 3.3 Knowledge Exploitation for Response Generation

We inject the generated five type of knowledge into dialogue historical context for better response generation. The integration of knowledge with the dialogue context is a critical factor, particularly considering the limited input length of language models. To address this, we adopt two approaches: utterance-level fine-grained knowledge injection (FgKI) for emotion, event, and world knowledge; dialogue-level coarse-grained knowledge injection (CgKI) for topic and persona.

**Utterance-Level Fine-Grained Knowledge Injection** We regard emotion, event, and general world knowledge as utterance-level fine-grained knowledge. And we use topic and persona knowledge as dialogue-level coarse-grained background knowledge $\mathcal{BK}$, denoted as "$[K_{tpc}; K_{psn}]$". We integrate emotion, event, and general world knowledge into the corresponding utterances of the dialogue. Hence, for the $x^{th}$ utterance ($x \in [1, t]$), the formulation of $U'_x$ is represented as "$[K^x_{emo}; K^x_{evt}; K^x_{wor}; U_x]$". Each knowledge type is associated with a specific conjunction to prevent confusion. To connect different rounds in the dialogue history, we use the labels "*System:*" and "*User:*". Additionally, we provide an instruction $\mathcal{I_G}$ for dialogue generation: "*Please consider the dialogue history, encompassing various types of knowledge such as emotion, topic, event, persona, and general world knowledge, to generate a response that exhibits rich diversity and coherency.*" We only apply this operation to the last two rounds of the dialogue. Therefore, the model output $S_t$ for this approach is:

$$S_t = \mathcal{LLM}([\mathcal{I_G}; \mathcal{BK}; U_1; S_1 \ldots U'_{t-1}; S_{t-1}; U'_t])$$

**Dialogue-level Coarse-Grained Knowledge Injection** We hypothesize that preceding sentences in the dialogue history have a lesser impact on response generation, as speakers tend to focus more on the most recent information. Therefore, in this approach, we specifically select the emotion, event, and general world knowledge from the last round to construct the background knowledge $\mathcal{BK}'$, denoted as "$[\mathcal{BK}; K^t_{emo}; K^t_{evt}; K^t_{wor}]$". We combine $\mathcal{BK}'$ with the dialogue historical context $\mathcal{U}$. Other settings, including conjunctions and instructions, remain consistent with the aforementioned method. Therefore, the model $S_t$ output at this time is:

$$S_t = \mathcal{LLM}([\mathcal{I_G}; \mathcal{BK}'; U_1; S_1 \ldots U_{t-1}; S_{t-1}; U_t])$$

## 4 Experimental Settings

### 4.1 Datasets

Since most of the current dialogue datasets focus on one or two kinds of dialogue knowledge, we conduct experiments on a variety of different types of datasets: EMPATHETICDIALOGUES (ED) (Rashkin et al., 2019), DAILYDIALOG (DD) (Li et al., 2017), PERSONACHAT (PC) (Zhang et al., 2018), and OPENDIALKG (ODKG) (Moon et al., 2019). The datasets used for the external knowledge sources and experiments share some common parts. However, please **NOTE** that: 1) the datasets we use for experiments do not contain any information besides the dialogue context; 2) the original train/dev/test split follows the original dataset, and the training set is utilized as external knowledge source, while the test set is utilized for experiments.

### 4.2 Baseline Methods

**General Large Language Models: OPT** (Zhang et al., 2022) is a suite of decoder-only pre-trained transformers with varying numbers of parameters, ranging from 125M to 175B. We use the 13B-parameter model of OPT. **ChatGLM-6B** (Zeng et al., 2022) is an open bilingual language model based on General Language Model (GLM) framework, with 6.2 billion parameters. ChatGLM uses technology similar to ChatGPT, and is optimized for Chinese QA and dialogue. **Flan-T5-XXL** is the backbone model of KnowEE. To demonstrate the effectiveness of our approach in enhancing response quality, we utilize it for direct generation.

**Pre-trained Dialogue Models: DialoGPT** (Zhang et al., 2019) is a fine-tuned GPT-2 (Radford et al., 2019) with Reddit comment data. We select

| Method | ED | | | DD | | | PC | | | ODKG | | |
|---|---|---|---|---|---|---|---|---|---|---|---|---|
| | PPL↓ | D-1 | D-2 | PPL↓ | D-1 | D-2 | PPL↓ | D-1 | D-2 | PPL↓ | D-1 | D-2 |
| OPT-13B | 34.86 | 9.01 | 46.16 | 19.86 | 13.39 | 50.02 | 24.31 | 10.12 | 32.36 | 24.62 | 15.66 | 49.47 |
| ChatGLM-6B | 30.12 | 3.85 | 23.61 | 19.30 | 8.10 | 38.77 | 22.92 | 4.21 | 22.52 | 33.75 | 8.87 | 32.42 |
| Flan-T5-XXL | 39.53 | 9.56 | 46.25 | 38.42 | 16.74 | 58.58 | 24.00 | 13.97 | 53.30 | 15.89 | 16.01 | 51.52 |
| DialoGPT | 21.54 | 6.69 | 24.03 | 17.91 | 10.84 | 34.56 | 27.71 | 9.54 | 30.91 | 16.01 | 7.91 | 25.05 |
| BlenderBot | 20.12 | 7.16 | 38.73 | 19.35 | 12.65 | 50.78 | **17.78** | 12.06 | 50.31 | 21.60 | 12.83 | 47.16 |
| FWP | 22.43 | 5.38 | 17.83 | 26.73 | 10.25 | 31.95 | 22.79 | 5.99 | 18.32 | 22.51 | 4.65 | 12.32 |
| FSB | 22.83 | 4.30 | 18.02 | 18.65 | 13.63 | 44.29 | 38.40 | 10.18 | 34.55 | 22.69 | 13.36 | 35.43 |
| MSDP | – | – | – | – | – | – | – | – | – | 24.41 | **20.71** | **63.69** |
| KnowEE-FgKI | 24.62 | 10.06 | **48.37** | **16.11** | 16.96 | 59.53 | 22.57 | **14.24** | 55.60 | **14.25** | 17.88 | 57.33 |
| KnowEE-CgKI | **19.04** | **11.95** | 52.13 | 16.68 | **17.48** | **61.36** | 22.50 | 14.13 | 54.74 | 15.83 | 17.74 | 56.18 |

Table 2: Automatic evaluation results of different methods and models on four datasets. The **bold** numbers in the results represent the best scores, whereas the underlined numbers indicate the second-best scores.

the 345M (best performance) for comparison. **Blenderbot** (Shuster et al., 2022) is a pre-trained conversational model which combines multiple models and techniques including GPT-2, BERT (Devlin et al., 2018), and Seq2Seq. We select Blenderbot-3B for comparison.

**Few-shot Learning Methods: FWP** (Zheng and Huang, 2021) is an approach to improve the dialogue generation task by learning continuous tokens to query the language model more efficiently. **FSB** (Madotto et al., 2021) is a chatbot which automatically selects the most appropriate conversational skill, queries different knowledge bases and uses the retrieved knowledge to generate a human-like response, all using only few dialogue examples. **MSDP** (Liu et al., 2022) is a few-shot framework that extracts general world knowledge from documents and employs it to generate responses via in-context learning. In MSDP, knowledge are obtained from Wikipedia (WoW) or Internet (WoI), which corresponds to general world knowledge in our work. Therefore, we only compare the performance of MSDP on the ODKG dataset.

### 4.3 Evaluation Metrics

**Automatic Metrics** Previous empirical studies reveal a significant gap between automatic metrics (e.g., BLEU (Papineni et al., 2002) and ROUGE (Lin, 2004)) and human judgments in evaluating dialogue generation (Liu et al., 2016). Nevertheless, there are reference-free metrics available that partially capture the quality of generated dialogues. Therefore, we utilize **Perplexity** (Jelinek et al., 1977) and **DISTINCT** (Li et al., 2016a) to evaluate the models. Perplexity (PPL) assesses the overall quality of the generation model, while DISTINCT (D-1 / D-2) measures the proportion of distinct

unigrams / bigrams in the generated outputs, indicating diversity. However, to ensure the integrity of our experimental results, we still represent the BLEU and ROUGE scores of various baselines on the ED dataset in Appendix A.

**Human Evaluation** We conduct a human evaluation on open-domain dialogue generation. We recruit university students to evaluate the quality of conversations. Considering the costs associated with human evaluation, **500** samples have been included for each baseline on each dataset. We follow up previous dialogue generation efforts (Yu et al., 2022; Li et al., 2022) and employ several general metrics to evaluate the dialogue quality : **Coherence** measures relevance to the dialogue context, **Informativeness** evaluates information provided, and **Fluency** checks grammatical accuracy.

Additionally, we use different metric for different dataset. For ED, we evaluate **Empathy**, measuring the match between the generated response and the speaker's emotion. For PC, we assess **Personality** consistency. For DD, we determine **Theme** adherence. For ODKG, we check for **Hallucination↓** and factual errors. Note that the Coherence, Informativeness, and Fluency scale is $[0, 1, 2, 3, 4]$, and Empathy, Personality and Theme scale is $[0, 1, 2]$, whose higher score indicates a better performance. Moreover, the scale of Hallucination is $[0, 1, 2]$, whose lower score indicates a better performance.

### 4.4 Implementation Details

We select Flan-T5-XXL (13B) as our backbone model. Flan-T5 (Chung et al., 2022) is a fine-tuned version model class of T5 (Raffel et al., 2019) that has been trained on a variety of datasets phrased as instructions. It has shown impressive performance on several benchmarks, demon-

| Method | ED | | | | DD | | | | ODKG | | | |
|---|---|---|---|---|---|---|---|---|---|---|---|---|
| | Cohe. | Info. | Flu. | Emp. | Cohe. | Info. | Flu. | The. | Cohe. | Info. | Flu. | Hall.↓ |
| OPT-13B | 1.56 | 1.94 | 1.13 | 0.69 | 1.34 | 1.07 | 1.11 | 0.56 | 2.01 | 1.42 | 1.69 | 1.03 |
| ChatGLM-6B | 2.11 | 2.08 | 2.14 | 1.02 | 2.49 | 2.61 | 2.09 | 1.23 | 1.38 | 1.50 | 1.33 | 1.28 |
| Flan-T5-XXL | 1.96 | 1.87 | 2.39 | 0.91 | 2.89 | 2.78 | 3.13 | 1.01 | 2.25 | 2.14 | 2.48 | 1.29 |
| DialoGPT | 1.92 | 1.19 | 1.81 | 0.77 | 2.20 | 1.79 | 2.39 | 1.28 | 2.73 | 2.27 | 2.76 | 1.18 |
| BlenderBot | 2.36 | 2.27 | **2.94** | 0.93 | 0.95 | 2.27 | 2.01 | 0.73 | 1.87 | 2.65 | **2.87** | 0.99 |
| FWP | 0.81 | 1.31 | 2.11 | 0.78 | 1.02 | 0.52 | 1.82 | 0.86 | 1.16 | 1.62 | 0.50 | 1.23 |
| FSB | 2.14 | 2.05 | 2.38 | 1.33 | 2.12 | 2.91 | 2.87 | 1.04 | 2.67 | 2.33 | 2.28 | 1.01 |
| MSDP | – | – | – | – | – | – | – | – | 2.92 | 3.01 | 1.54 | 0.88 |
| KnowEE-FgKI | 2.57 | **2.86** | 2.71 | 1.43 | 3.09 | **3.14** | 2.63 | 1.41 | **3.09** | 2.83 | 2.43 | 0.96 |
| KnowEE-CgKI | **2.67** | 2.75 | 2.74 | **1.57** | **3.19** | 2.94 | **3.59** | **1.44** | 3.01 | **3.13** | 2.74 | **0.83** |

Table 3: Human evaluation results of different methods and models on ED, DD and ODKG, where Cohe., Info., Flu., Emp., The., and Hall.↓ are the abbreviations corresponding to *Coherence*, *Informativeness*, *Fluency*, *Empathy*, *Theme* and *Hallucination*.

| Method | PPL | D-2 | Cohe. | Info. | Flu. | Emp. |
|---|---|---|---|---|---|---|
| KnowEE | 19.04 | 52.13 | 2.67 | 2.75 | 2.74 | 1.57 |
| w/o Emo | 19.25 | 45.23 | 2.56 | 2.31 | 2.68 | 1.05 |
| w/o Tpc | 18.67 | 45.28 | 2.51 | 2.33 | 2.69 | 1.35 |
| w/o Psn | 19.34 | 45.10 | 2.60 | 2.62 | 2.73 | 1.36 |
| w/o Evt | 18.53 | 44.35 | 2.59 | 2.66 | 2.62 | 1.39 |
| w/o Wor | 18.10 | 42.07 | 2.61 | 2.68 | 2.72 | 1.46 |

Table 4: Ablation study of our proposed framework. The abbreviation **w/o** is used to indicate that the LLM generates responses without using the corresponding type of knowledge.

strating strong zero-shot, few-shot, and Chain-of-Thought (CoT) (Wei et al., 2022) abilities. Flan-T5-XXL is the largest released checkpoint of this model, boasting a parameter volume of 13B. We leave LLaMA (Touvron et al., 2023) based models such as Alpaca (Taori et al., 2023) and Vicuna (Chiang et al., 2023) in the future research work.

## 5 Results and Analysis

### 5.1 Open-domain Dialogue Generation

We conduct automatic and human evaluations to compare our proposed framework, KnowEE, with the baselines mentioned earlier, using four datasets.

**Automatic Evaluation Results**    As shown in Table 2, the experimental results show that KnowEE overall outperforms the baselines in perplexity and diversity across all datasets. Only Blenderbot slightly outperforms our framework with different knowledge injection method in perplexity for ED and PC. Besides, MSDP outperforms our framework in terms of diversity for ODKG. However, KnowEE surpasses all other baselines in overall

performance. As KnowEE is capable of securing the **Top 3** positions in all automatic metric scores across all datasets. This shows that our method can achieve better diversity while ensuring lower perplexity.

**Human Evaluation Results**    The results of all models or methods for ED, DD, and ODKG are presented in Table 3. For brevity, the remaining results can be found in Appendix B. These tables reveal that KnowEE achieves a prominent position in human evaluation scores, indicating the following trends:

First, KnowEE outperforms existing approaches in human evaluation metrics, showcasing competitive performance in coherence, fluency, and informativeness. The results emphasize the effectiveness of leveraging LLMs for open-domain dialogue tasks via pre-response knowledge generation using well-designed prompt patterns.

Second, compared to large-scale dialogue models like ChatGLM and pre-trained dialogue models such as DialoGPT, KnowEE exhibits superior coherence and informativeness, highlighting the effectiveness of its knowledge generation and injection mechanism. Additionally, dialogue models with specific architectures and adequate training consistently excel in terms of fluency, supporting the benefits of pre-trained dialogue models in generating dialogue responses, as confirmed by empirical analysis of BlenderBot and DialoGPT.

Besides general human evaluation metrics, our main focus is on whether the model incorporates implicit knowledge, such as emotions and themes, from the dialogue history during generation. To achieve this, we develop specific metrics (Empathy,

| Method | ED | | | | | | | ODKG | | | | | | |
|---|---|---|---|---|---|---|---|---|---|---|---|---|---|---|
| | PPL↓ | D-1 | D-2 | Cohe. | Info. | Flu. | Emp. | PPL↓ | D-1 | D-2 | Cohe. | Info. | Flu. | Hall.↓ |
| **ChatGLM** | 14.13 | 6.03 | 19.34 | 2.11 | 2.08 | 2.14 | 1.02 | 15.98 | 8.89 | 33.21 | 1.38 | 1.50 | 1.33 | 1.28 |
| w FgKI | 21.85 | 6.74 | 27.52 | 2.09 | *2.34* | *2.22* | 1.16 | 16.02 | 5.82 | 20.53 | *2.25* | 2.94 | 1.54 | *0.65* |
| w CgKI | *13.16* | *7.47* | *31.48* | 2.26 | 2.18 | 2.14 | *1.29* | *14.49* | *13.89* | *43.48* | 2.20 | *3.18* | *1.88* | 0.89 |
| **ChatGPT** | 16.43 | 16.85 | 50.04 | 3.51 | 3.50 | **4.00** | 1.63 | 20.38 | 22.08 | 50.58 | 3.57 | 3.69 | **4.00** | 0.44 |
| w FgKI | 10.63 | 19.13 | 58.48 | 3.63 | 3.25 | **4.00** | 1.66 | 13.96 | **26.56** | 60.90 | 3.57 | **3.80** | **4.00** | 0.43 |
| w CgKI | **10.37** | **19.15** | **61.02** | **3.65** | **3.60** | **4.00** | **1.68** | **13.86** | 26.46 | **62.23** | **3.69** | 3.79 | **4.00** | **0.40** |

| Method | DD | | | | | | | PC | | | | | | |
|---|---|---|---|---|---|---|---|---|---|---|---|---|---|---|
| | PPL↓ | D-1 | D-2 | Cohe. | Info. | Flu. | The. | PPL↓ | D-1 | D-2 | Cohe. | Info. | Flu. | Per. |
| **ChatGLM** | 19.30 | 8.10 | 38.77 | 2.49 | 2.61 | 2.09 | 1.23 | 22.92 | 4.21 | 22.52 | 2.11 | 2.71 | 1.29 | 1.30 |
| w FgKI | 20.51 | 5.13 | 20.58 | 2.63 | 2.71 | 2.11 | 1.33 | 24.79 | 4.75 | 20.37 | 1.38 | 2.11 | 0.94 | 0.71 |
| w CgKI | *19.09* | *10.84* | *40.50* | 2.69 | *3.01* | *2.60* | *1.39* | *15.61* | *7.34* | *32.21* | 2.94 | 2.90 | *1.50* | *1.31* |
| **ChatGPT** | 20.30 | 19.37 | 49.70 | 3.86 | 3.72 | **4.00** | 1.57 | 19.88 | 18.74 | 48.69 | 3.56 | 3.17 | **4.00** | 1.22 |
| w FgKI | **13.68** | 23.61 | 62.23 | 3.88 | **3.73** | **4.00** | 1.90 | 12.63 | 23.04 | 61.32 | **3.60** | 3.19 | **4.00** | 1.36 |
| w CgKI | 14.18 | **23.89** | **63.49** | **3.90** | 3.72 | **4.00** | **1.93** | **12.47** | **23.90** | **64.57** | 3.59 | **3.23** | **4.00** | **1.38** |

Table 5: Generalization ability study of our proposed framework. **w FgKI** and **w CgKI** represent LLM generates responses using KnowEE-FgKI and KnowEE-CgKI respectively.

Theme, Hallucination, and Personality) for each dataset. Table 3 presents the results. Surprisingly, models like BlenderBot, which excel in coherence, informativeness, and fluency, struggle with understanding dialogue knowledge accurately, leading to lower scores on relevant metrics. Case analysis shows that they often deviate from the conversation theme, character personality, or express incorrect emotions, occasionally containing factual errors. In contrast, our framework consistently ranks highest across all relevant indicators for all datasets. Notably, KnowEE with coarse-grained injection generally outperforms fine-grained method, possibly due to the superior comprehension abilities of Flan-T5-XXL in processing input.

Last but not least, compared to MSDP, our approach excels in terms of coherence and informativeness. Besides, MSDP even scores lower in fluency compared to Flan-T5-XXL in zero-shot scenarios. Through case analysis, we find that Flan-T5-XXL with MSDP has a probability of copying content from generated knowledge during text generation, which significantly reduces the fluency of results. In contrast, our method considers various types of knowledge during text generation, which avoids the scenario where the model is overly influenced by a single type of knowledge, and enables us to mitigate this issue.

## 5.2 Ablation Study

The aim of this study is to investigate the impact of different dialogue knowledge types on response generation. Ablation experiment on ED is conducted, removing one knowledge type at a time (**w/o Emo**, **w/o Tpc**, **w/o Psn**, **w/o Evt**, w/o Wor). Evaluation results in Table 4 reveal that removing any knowledge type reduces perplexity, diversity, relevance, fluency, informativeness, and emotional matching in responses. Emotional knowledge (**w/o Emo**) has the greatest impact, indicating its significant role in understanding dialogue history and affecting response quality. Results and analysis demonstrate that the dialogue knowledge obtained from LLMs contributes significantly to generating responses.

## 5.3 Generalization Ability Analysis

We compare the performance of KnowEE with different backbone models to evaluate the generalization ability of our framework. We select two instruction-tuned models with larger scale and parameters, specifically ChatGPT and ChatGLM. ChatGPT is not open-source and can only be accessed through API from OpenAI [2] for inference.

From Table 5, we observe that although ChatGPT and ChatGLM exhibit a strong dialogue response generation ability, the scores for all automatic and human evaluation metrics of responses generated by these models using KnowEE framework are consistently higher or equal to those directly using them for inference. This demonstrates that our proposed framework is still effective for models with larger scale, stronger knowledge re-

[2]https://openai.com/

serve, and understanding ability, and it is agnostic to the type of base model.

## 5.4 Case Analysis

We conduct a case study to further prove the advantages of our proposed framework. We select several test generation results from the four datasets, with the expectation of observing improvements in the performance of LLMs in dialogue response tasks with the use of our framework. We present details in Appendix C due to space limitation.

## 6 Conclusion

In this paper, we study the task of open-domain knowledge generation and identifies challenges of lacking various types of knowledge from diverse sources. To address these challenges, we identify five key type of dialogue knowledge, and propose a framework called KnowEE that explores multi-source multi-type knowledge from large language models using external datasets, and exploits the knowledge in fine-grained and coarse-grained manners for response generation. It explores knowledge from LLMs and injects knowledge into dialogue context to generate final responses. Experiments on four dialogue datasets show that KnowEE enhances LLM's understanding of dialogue context and improves generated responses in terms of coherence, informativeness, and fluency. Moreover, KnowEE exhibits robust generalization capabilities, making it applicable to multiple LLMs.

In our future endeavors, we intend to refine the dialogue knowledge further and explore the untapped potential of large language models in tasks that extend beyond dialogue response.

## Limitations

The limitations of our framework mainly come from the disadvantages of using large language models. First of all, most of the large language models that work well are not open source or free. This makes it difficult to conduct batch experiments or daily use on it. Next, a small number of open-source models require a lot of GPU resources when used, which is a difficult problem for quite many researchers, such as students.

## Ethics Statement

We honor and support the ACL code of Ethics. Our proposed framework KnowEE aims to generate friendly, high-quality, informative, and coherent responses. The interaction and assistance process do not involve any bias towards to the participants. All datasets used in this work are from previously published works, and in our view, do not have any attached privacy or ethical issues.

## Acknowledgements

This research is supported by the National Natural Science Foundation of China (No.62106105, No.62272274), the CCF-Baidu Open Fund (No.CCF-Baidu202307), the CCF-Tencent Open Research Fund (No.RAGR20220122), the CCF-Zhipu AI Large Model Fund (No.CCF-Zhipu202315), the Fundamental Research Funds for the Central Universities (No.NJ2023032), the Scientific Research Starting Foundation of Nanjing University of Aeronautics and Astronautics (No.YQR21022), and the High Performance Computing Platform of Nanjing University of Aeronautics and Astronautics.

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

# A  Automatic Evaluation Results of BLEU and ROUGE

The metrics based on N-gram content match, such as BLEU and ROUGE have already gained consensus as inappropriate for evaluating text generation, particularly dialogue generation. However, In order to further illustrate the effectiveness of our framework, we employ these metrics to evaluate certain methods from Table 2 and Table 3 that obtain higher scores in both human and automatic evaluations. Taking the ED dataset as an example, the BLEU (BLEU-1, 2, 3, 4) and ROUGE (ROUGE-1, 2, L) scores for various methods are presented in Table 6.

The results indicate that even considering only the n-grams match of response content, all metric scores of KnowEE-FgKI and KnowEE-CgKI exceed all baselines. This further supports the conclusion drawn from our previous experimental analysis, which is the effectiveness of our framework.

# B  Human Evaluation Results of DD

As shown in Table 7, our proposed framework can achieve an overall leading position, compared to strong baselines on DD, and coarse-grained method has a slight advantage over fine-grained method.

| Method | BLEU-1 | BLEU-2 | BLEU-3 | BLEU-4 | ROUGE-1 | ROUGE-2 | ROUGE-L |
|---|---|---|---|---|---|---|---|
| ChatGLM-6B | 5.45 | 1.69 | 0.63 | 0.25 | 12.98 | 1.27 | 10.34 |
| Flan-T5-XXL | 7.61 | 2.92 | 1.30 | 0.62 | 11.58 | 1.77 | 10.31 |
| DialoGPT | 5.16 | 2.07 | 0.99 | 0.54 | 13.23 | _2.01_ | **12.31** |
| BlenderBot | 7.51 | 2.03 | 0.71 | 0.30 | **13.43** | 1.35 | 10.98 |
| FSB | 7.24 | 2.15 | 0.92 | 0.49 | 12.75 | 1.43 | 11.39 |
| KnowEE-FgKI | _10.78_ | _3.81_ | _1.68_ | _0.93_ | 12.46 | 1.66 | 10.73 |
| KnowEE-CgKI | **11.84** | **4.40** | **2.13** | **1.27** | _13.40_ | **2.02** | _11.47_ |

Table 6: BLEU and ROUGE results of KnowEE and several strong baselines on ED.

| Models | Cohe. | Info. | Flu. | Per. |
|---|---|---|---|---|
| OPT-13B | 1.37 | 1.41 | 1.02 | 1.05 |
| ChatGLM-6B | 2.11 | 2.71 | 1.29 | 1.30 |
| Flan-T5-XXL | **2.53** | 2.56 | 2.77 | 1.21 |
| DialoGPT | 2.03 | 2.16 | 2.24 | 1.22 |
| BlenderBot | 2.03 | 2.90 | 2.96 | 0.63 |
| FWP | 0.53 | 1.86 | 1.94 | 0.53 |
| FSB | 2.27 | 2.93 | 2.79 | 1.30 |
| KnowEE-FgKI | 2.36 | **3.01** | _3.14_ | _1.53_ |
| KnowEE-CgKI | _2.41_ | _2.99_ | **3.27** | **1.56** |

Table 7: Human evaluation results of DD, where Per. are the abbreviations corresponding to *Personality*.

These results match our previous analysis in section 5.1.

## C Case Analysis

We conduct a case study to further prove the advantages of our proposed framework. We select several test examples from four datasets and use Flan-T5-XXL (the backbone model of KnowEE), KnowEE-FgKI, and KnowEE-CgKI for response generation, with the expectation of observing improvements in the performance of LLMs in dialogue response tasks with the use of our framework.

As shown in Table 8, in the first case, the dialogue history involves the user's experience of visiting an orphan and learning from the kid. However, Flan-T5-XXL fails to recognize the user's emotions and gives a bland response. While with the help of pre-generated emotion and event knowledge, the model finally produces a response expressing wonder and curiosity about the user's surreal moment, which directs the user to continue the conversation. Similarly, in the second case, the dialogue history focuses on a romance novel, and the user is seeking recommendations. Flan-T5-XXL, however, recommends a historical novel. By generating general world knowledge about romance novels and event

knowledge about the user's intention, the model combines the dialogue history and dialogue knowledge, producing different results. Notably, Flan-T5-XXL with the coarse-grained method provided the appropriate book recommendation.

As shown in Table 9. In the first case of DD, the user inquires about the price of a map. Flan-T5-XXL fails to comprehend the dialogue and provides a meaningless generic response. However, by explicitly incorporating event and topic knowledge to discern the user's intent, our framework generates relevant responses. Among them, only the response from KnowEE-FgKI comes closest to the ground truth, whereas the response from KnowEE-CgKI is deemed unreasonable. In the second case of PC, the user assumes the role of the author, while the system acts as a secretary. The response generated by Flan-T5-XXL is unhelpful and highly irrelevant. In contrast, our framework is capable of producing responses that align with the user's personality and are contextually relevant.

The results and analysis of the case study demonstrate the importance of injecting dialogue knowledge into the input of LLMs before generating responses. Our proposed framework effectively stimulates the storage knowledge of LLMs about dialogue history, resulting in higher-quality dialogue responses.

| Empathetic Dialogues | |
|---|---|
| **Dialogue History** | *User*: I visited an orphan once and it was such a surreal moment. It made me realise how thankful I should be for a family. (**Thankful**) |
| | *System*: That must have been and intense and emotional moment for you. |
| | *User*: Yes it definitely was. I went with intentions to teach the kid a few things, but I ended up learning lots more from him. (**Grateful**) |
| **Key Knowledge** | *Emotion*: grateful. *Event*: The user described a surreal moment; The user ended up learning lots more from the kids. |
| **Ground Truth** | What a surreal moment. I always believe kids have a lot to teach us. |
| **Flan-T5-XXL** | Well thats what happens a lot of times. (**Neutral**) |
| **KnowEE-FgKI** | *Amazing*! How old was he when you met him? (**Wonder and curiosity**) |
| **KnowEE-CgKI** | What an *amazing* experience that must have been. (**Wonder and curiosity**) |

| Open Dialog KG | |
|---|---|
| **Dialogue History** | *User*: I like the Author Jane Austen. Do you have any recommendations of books like hers? |
| | *System*: Sense and Sensibility |
| | *User*: Can you tell me more about that book? What genre is it? |
| | *System*: Its a Romance novel |
| | *User*: Oh! Sounds great. I like **Romance novels**. Can you recommend any others? |
| **Key Knowledge** | *General World*: Romance novel is a fictional genre of book centered around a love story. *Event*: The user wants to get some recommendations. |
| **Ground Truth** | From prada to nada |
| **Flan-T5-XXL** | *The Kingmaker's Daughter* is another one I can recommend. (**Historical novel**) |
| **KnowEE-FgKI** | *Three Men in a Boat*, I'd say that's another good one. (**Humor novel**) |
| **KnowEE-CgKI** | I have a book in my hands right now. It is called *Twilight*. (**Romance novel**) |

Table 8: Two cases of generated responses by Flan-T5-XXL and our framework. In the case of ED, we annotate the user's utterances with emotional labels. In the case of ODKG, we bold the key words that are decisive for generating the response. Specifically, we highlight words related to dialogue knowledge in blue and words relevant to the predicted content in red.

| | **DailyDialog** |
|---|---|
| **Dialogue History** | *User*: Do you have **maps** of downtown area? |
| | *System*: Yes, here you are. |
| | *User*: **How much** is it? |
| **Key Knowledge** | *Topic*: Ask for directions. *Event*: The user asks for maps of downtown area.; The user wants to buy something. |
| **Ground Truth** | It's *free* of charge. (**Free gift**) |
| **Flan-T5-XXL** | Thank you. |
| **KnowEE-FgKI** | It's *ninety* dollars. (**Charge ninety dollars**) |
| **KnowEE-CgKI** | It's *free*. Is that ok? (**Free gift**) |

| | **Persona Chat** |
|---|---|
| **Dialogue History** | *User*: What do you do for a living? |
| | *System*: I am a **secretary** like my mom hard living, and you? |
| | *User*: I am an **author**. I **write short stories**. |
| **Key Knowledge** | *Persona*: The user is an author; The system is an secretary. |
| **Ground Truth** | Do you want to *write a story* about a major meat eating bodybuilder? |
| **Flan-T5-XXL** | It is hard living, its very lonely. |
| **KnowEE-FgKI** | I *write short stories* too. Do you have an agent? |
| **KnowEE-CgKI** | I *write short stories* too. Do you mind if I get an agent to represent me? |

Table 9: Two cases of generated responses by Flan-T5-XXL and our framework. We bold the key words that are decisive for generating the response. We highlight words related to dialogue knowledge in blue and words relevant to the predicted content in red.