# OpenReview forum: "Multi-Source Multi-Type Knowledge Exploration and Exploitation for Dialogue Generation"
_EMNLP/2023/Conference — EMNLP 2023 Main_

### Official Review · Reviewer_keZg · 2023-07-31

**Soundness:** 4

**Excitement:**

3: Ambivalent: It has merits (e.g., it reports state-of-the-art results, the idea is nice), but there are key weaknesses (e.g., it describes incremental work), and it can significantly benefit from another round of revision. However, I won't object to accepting it if my co-reviewers champion it.

**Paper Topic And Main Contributions:**

The paper presents a framework, KnowEE, for retrieving knowledge for open-domain dialogue generation from LLMs and injecting that knowledge into LLM prompts to generate dialogue responses. The authors argue that training knowledge retrievers is both computationally expensive and brittle, with poor generalization to different knowledge sources. They also note that LLMs contain a large amount of knowledge from their pretraining but are often unable to retrieve and use that knowledge when prompted with only the dialogue history.

The authors propose to retrieve LLM knowledge by identifying the top n most similar content-knowledge pairs from one of five knowledge datasets representing different knowledge types (eg. emotion, personal, world knowledge); the pairs are used as in-context learning examples to prompt the LLM to retrieve knowledge for the current utterance or dialogue. The knowledge is then injected into the response generation prompt in one of two ways: at the utterance level, where knowledge is retrieved for the last two user utterances in the dialogue history, or at the dialogue level, where the last user utterance is used as a proxy to retrieve knowledge for the whole dialogue.

The experimental results include two automatic metrics, perplexity and DISTINCT, and a human evaluation including custom categories specific to each knowledge type. The authors find that the proposed KnowEE approach outperforms existing non-knowledge-based systems, including both LLMs and pretrained dialogue models, as well as two other few-shot learning, knowledge-based models.

**Questions For The Authors:**

The use of "multi-source" for this approach is slightly confusing. Multi-source would usually refer to the five different knowledge datasets, but the proposed approach does not actually retrieve knowledge from these sources. There is only one source, the LLM, and the five different datasets are used for in-context learning prompts to retrieve the LLM knowledge.

Can you explain what the difference is between Fine-Grained Knowledge Injection and Coarse-Grained Knowledge Injection? From the descriptions in Section 3.3, it seems that both approaches have access to K^t_emo, K^t_evt, and K^t_wor, although this knowledge appears in a different linear position in the prompt. Does the position matter? There is nothing in the prompt that instructs the LLM to treat the knowledge differently based on where it appears. Section 5.1 mentions that coarse-grained outperforms fine-trained "possibly due to the superior comprehension abilities of Flan-T5-XXL in processing input," which I do not understand because Flan-T5-XXL is used as the backbone for both approaches; is this about the positioning?

As far as I can tell, only the K^t-1 knowledge is missing from the coarse-grained approach. If this is the case, then "fine-grained" and "coarse-grained" are confusing names for these two approaches, since they both use utterance-level knowledge. From the name, I would have expected the coarse-grained approach to use the entire dialogue context as the query for all knowledge types.

How many samples are included in the human evaluation? I am concerned because the metrics used for automatic evaluation, perplexity and DISTINCT, evaluate the generation quality, not the content; BLEU and ROUGE have their limitations, but they would have evaluated the response content, which I would argue is more important than text quality for a knowledge-based approach. We must depend entirely on the human evaluation's Coherence, Informativeness, and knowledge-specific categories to evaluate the response content, but human evaluations are usually small compared to the overall test set size, especially with as many comparison methods as are used in these experiments.
Update: This concern was addressed in the author rebuttal.

It would also be helpful to see the evaluation results for a traditional, trained knowledge-retrieval dialogue system as a point of comparison. While the proposed approach is shown to outperform non-knowledge-based approaches and other few-shot learning-based approaches, the introduction section motivates the current work as an alternative to a trained knowledge retriever.

**Reasons To Accept:**

The proposed approach is well-motivated, clearly explained, and outperforms almost all comparison systems on almost all metrics. The framework is agnostic to the choice of underlying LLM, and the authors demonstrate that it can be used to improve dialogue response generation performance on several LLMs.

**Reasons To Reject:**

The two proposed knowledge injection approaches, utterance-based fine-grained and dialogue-based coarse-grained, seem very similar to me. From the description in the text, they seem to differ only in whether or not the penultimate user utterance is used. The experimental results show that coarse-grained outperforms fine-grained, but I am not sure what the takeaway lesson is because they are so similar to each other.

Additionally, I think the evaluation would be stronger with the inclusion of content-based automatic metrics like BLEU and ROUGE, as well as additional comparison systems representing the trained knowledge retrieval approach.
Update: This concern was addressed in the author rebuttal.

(See questions below for more deatils.)

**Reproducibility:**

4: Could mostly reproduce the results, but there may be some variation because of sample variance or minor variations in their interpretation of the protocol or method.

**Reviewer Confidence:**

3: Pretty sure, but there's a chance I missed something. Although I have a good feel for this area in general, I did not carefully check the paper's details, e.g., the math, experimental design, or novelty.

**Typos Grammar Style And Presentation Improvements:**

Typos:
Introduction, Line 100: person -> persons
Introduciton, Line 134: propsoed -> proposed
Experimental Settings, Datasets, Line 388: "... is followed the original dataset" -> "...is followed"
Experimental Settings, Datasets, Line 389: trainset -> training set
Experimental Settings, Datasets, Line 390: testset -> test set
Results and Analysis, Human Evaluation Results, Line 533: fin-grained -> fine-grained

---

> ### Author Rebuttal · Authors · 2023-08-27
>
> We sincerely appreciate your valuable feedback on our paper. Your insights are of utmost importance to us, and we would like to provide comprehensive explanations addressing the concerns you have raised.
>
> **Question 1: The use of "multi-source" for this approach is slightly confusing.**
>
> The term "multi-source" here indeed refers to the utilization of five external datasets for in-context learning. Our rationale for this choice stems from the belief that LLMs have already assimilated substantial knowledge from diverse sources during pre-training, effectively rendering them aggregations of various knowledge sources. Our concept revolves around leveraging multiple datasets to stimulate the pertinent knowledge stored in LLMs. We apologize for any confusion arising from this expression and acknowledge the possibility of amending this term.
>
> **Question 2: Explain the difference between Fine-Grained Knowledge Injection and Coarse-Grained Knowledge Injection.**
>
> Due to the input token length limitations of Flan-T5-XXL (2048), as dialogues extend over multiple rounds, they frequently surpass this limit. Consequently, our endeavor aims to reduce input length while circumventing knowledge attrition. Initially, we adopted a coarse-grained approach: the concatenation of knowledge corresponding to each utterance in dialogue history. Evidently, this strategy precipitated rapid inflation of the input token length. This prompted us to opt exclusively for the last round's emotion, event, and general world knowledge. We observed marginal disparities in the results between these two methods, with the latter even outperforming the former in most instances. This rationale underpins our exposition of the two knowledge injection methods in paper. As to why the latter method displayed superior performance, we attribute it to the inclusion of unadulterated dialogue history. This holistic dialogue contextual comprehension potentially enhances response generation.
>
> We regret any confusion stemming from our nomenclature of the two knowledge injection methods. Concurrently, we appreciate your interpretation of the coarse-grained approach: utilizing the entire context of the dialogue as a query for all types of knowledge, which closely aligns with the concept of "coarse-grained". We are inclined to refine the description of Coarse-Grained Knowledge Injection method and intend to employ the entire contextual history as a query for knowledge generation in subsequent work.
>
> **Question 3: Num of samples that are included in the human evaluation. Missing metrics like BLEU and ROUGE to evaluate the response content.**
>
> Considering the costs associated with human evaluation, **500** samples have been included for each method across each dataset. The metrics based on N-gram content matching, such as Bleu and Rouge that you mentioned, have already gained consensus as inappropriate for evaluating text generation, particularly dialogue generation. As a result, we did not initially consider incorporating such metrics. In order to address your concerns, we employ these metrics to evaluate certain methods from Table 2 and Table 3 of the paper that obtain higher scores in both human and automatic evaluations. Taking the ED dataset as an example, the BLEU (*BLEU-1, 2, 3, 4*) and ROUGE (*ROUGE-1, 2, L*)  scores for various methods are presented below:
>
> | Method      | BLEU-1    | BLEU-2   | BLEU-3   | BELU-4   | ROUGE-1   | ROUGE-2  | ROUGE-L   |
> | ----------- | --------- | -------- | -------- | -------- | --------- | -------- | --------- |
> | ChatGLM-6B  | 5.45      | 1.69     | 0.63     | 0.25     | 12.98     | 1.27     | 10.34     |
> | Flan-T5-XXL | 7.61      | 2.92     | 1.30     | 0.62     | 11.58     | 1.77     | 10.31     |
> | DialoGPT    | 5.16      | 2.07     | 0.99     | 0.54     | 13.23     | 2.01     | **12.31** |
> | BlenderBot  | 7.51      | 2.03     | 0.71     | 0.30     | **13.43** | 1.35     | 10.98     |
> | FSB         | 7.24      | 2.15     | 0.92     | 0.49     | 12.75     | 1.43     | 11.39     |
> | KnowEE-FgKI | 10.78     | 3.81     | 1.68     | 0.93     | 12.46     | 1.66     | 10.73     |
> | KnowEE-CgKI | **11.84** | **4.40** | **2.13** | **1.27** | 13.40     | **2.02** | 11.47     |
>
> The results indicate that our approach continues to maintain a overall leading position in terms of the response content. We will incorporate these findings into the paper to enhance the robustness of our experiments. Thank you once again for your invaluable suggestions.
>
> **Question 4: It would also be helpful to see the evaluation results for a traditional, trained knowledge-retrieval dialogue system as a point of comparison.**
>
> We extend our sincere gratitude for your valuable recommendations concerning our experimental setup. Due to time constraints, we are unable to report the result of an appropriate traditional, trained knowledge-retrieval dialogue system as a baseline during the rebuttal period. Consequently, we have deferred this aspect of the work to a later phase. We are committed to incorporating this experiment part in the next version of the paper.
>
> **Question 5: Typos Grammar Style And Presentation Improvements**
>
> We apologize for any spelling and grammatical errors that may have detracted from your reading experience. We will rectify the issues you've raised and meticulously review the entire paper to prevent the existence of similar problems.
>
> Finally, we are committed to expanding and refining our research based on your suggestions. Your feedback will be carefully considered, with the aim of further enhancing the significance and value of our study.

---

### Official Review · Reviewer_eFw1 · 2023-08-04

**Soundness:** 4

**Excitement:**

4: Strong: This paper deepens the understanding of some phenomenon or lowers the barriers to an existing research direction.

**Paper Topic And Main Contributions:**

The paper addresses the challenge in generating multi-turn dialogues by leveraging diverse knowledge from various sources. Existing methods focus on specific dialogue knowledge and corresponding datasets, limiting generalization and resource requirements. The proposed "KnowEE" framework employs large-scale language models (LLMs) to overcome this. It consists of two phases: First, it extracts relevant samples from five knowledge-rich datasets to create prompts for generating knowledge. Second, it integrates this knowledge into ongoing dialogues at different levels of detail and feeds it into LLMs to produce coherent, informative responses. Both automated and manual evaluations confirm the framework's effectiveness in utilizing multi-source, multi-type knowledge for generating fluent and meaningful dialogue responses.

**Reasons To Accept:**

1. Novel approach for multi-turn dialogue generation. By leveraging large-scale language models and multiple external datasets, the proposed framework extracts and injects relevant knowledge into dialogues. This approach goes beyond existing methods that focus on specific datasets, showcasing innovation in utilizing LLMs for knowledge integration in dialogues.
2. Effective validation through evaluation. The paper presents both automatic and manual evaluation results that validate the effectiveness of the proposed framework.

**Reasons To Reject:**

Lack the generalization analysis to real-world scenarios. While the proposed framework shows promise in generating coherent responses, the paper lacks a discussion on how well it would generalize to real-world scenarios and diverse user interactions.

**Reproducibility:**

4: Could mostly reproduce the results, but there may be some variation because of sample variance or minor variations in their interpretation of the protocol or method.

**Reviewer Confidence:**

4: Quite sure. I tried to check the important points carefully. It's unlikely, though conceivable, that I missed something that should affect my ratings.

---

> ### Author Rebuttal · Authors · 2023-08-27
>
> We sincerely appreciate your valuable feedback on our paper. Your insights are of utmost importance to us, and we would like to provide comprehensive explanations addressing the concerns you have raised.
>
> **Question: Lack a discussion on how well KnowEE would generalize to real-world scenarios and diverse user interactions.**
>
> First and foremost, we greatly appreciate your recognition of our work. One of the central objectives of this study is to apply LLMs to downstream generation tasks in a concise and effective manner. In our future endeavors, we will employ more powerful and innovative LLMs such as LLaMA and Vicuna. Additionally, we will investigate the application of KnowEE in the context of smaller-scale models.
>
> Secondly, we acknowledge the absence of generalization analysis to real-world scenarios in our work. Therefore, we are establishing an authentic dialogue system. We will select 500 dialogue utterances randomly from four datasets to serve as initial sentences. For each of these sentences, we will engage university students for real conversational interactions. Specifically, after users input the start sentence, a three-round dialogue will ensue, and users will give the rating about the reponses. Due to time constraints, we are unable to report this result during the rebuttal period. Consequently, we have deferred this aspect of the work to a later phase. We will incorporate this experiment part in the next version of the paper.
>
> We are committed to expanding and refining our research based on your suggestions. Your feedback will be carefully considered, with the aim of further enhancing the significance and value of our study.

---

### Official Review · Reviewer_Q3qA · 2023-08-04

**Soundness:** 3

**Excitement:**

3: Ambivalent: It has merits (e.g., it reports state-of-the-art results, the idea is nice), but there are key weaknesses (e.g., it describes incremental work), and it can significantly benefit from another round of revision. However, I won't object to accepting it if my co-reviewers champion it.

**Missing References:**

1. ACL2022 Multi-Stage Prompting for Knowledgeable Dialogue Generation

**Paper Topic And Main Contributions:**

This paper introduces a new method of KnowEE to flexibly integrate multiple knowledge sources into dialogue generation. Specifically, the new method includes two steps: multi-source in-context learning and fine/coarse-grained knowledge injection for response generation. At last, the method is tested on different dialogue datasets and proven to achieve noticeable improvements.

**Reasons To Accept:**

1. The paper is well-organized, contains enough information in a limited number of pages, and is easy to understand.
2. The paper evaluates different variants of the method KnowEE-FgKI and KnowEE-CgKI, and they generally achieve better results than baselines in both automatic and human evaluation. The experiments in the paper are extensive and convincing. The chosen evaluation metrics are presentative, and ablation experiments show the efficacy of the KnowEE's design choices.

**Reasons To Reject:**

1. This proposed method seems to be the multi-source adoption of an existing method MSDP (https://arxiv.org/abs/2203.08745).  It seems to be incremental that the new method just adds multiple knowledge sources and in-context learning with demonstration retrieval (such as https://arxiv.org/pdf/2305.04320.pdf). Thus, the work lacks core novelty.
2. There is no ablation experiment comparing MSDP'performance with in-context learning and that only with zero-shot. There is no experimental support for the claim that in-context learning actually helps LLMs in inferring emotion, topic, and event knowledge.

**Reproducibility:**

3: Could reproduce the results with some difficulty. The settings of parameters are underspecified or subjectively determined; the training/evaluation data are not widely available.

**Reviewer Confidence:**

4: Quite sure. I tried to check the important points carefully. It's unlikely, though conceivable, that I missed something that should affect my ratings.

---

> ### Author Rebuttal · Authors · 2023-08-27
>
> We sincerely appreciate your valuable feedback on our paper. Your insights are of utmost importance to us, and we would like to provide comprehensive explanations addressing the concerns you have raised.
>
> **Question 1: This proposed method seems to be the multi-source adoption of an existing method MSDP.**
>
> We have thoroughly read the paper you mentioned: "MSDP"[1]. In this work, although a similar framework of generating knowledge first and then generating responses is employed, there exist distinctions between our approach and theirs:
>
> - In their experiments, knowledge are obtained from Wikipedia (WoW) or Internet (WoI), which corresponds to general world knowledge in our work. Conversely, our work delves into the more types of dialogue knowledge from diverse sources, such as topic, emotion, etc. The applicability of our framework is notably more extensive, as it can be employed in real-world open-domain dialogue systems due to its rich knowledge sources and types. By inputting dialogue history, our method explores and utilizes LLMs' knowledge to generate high-quality responses. In contrast, MSDP is confined to question answering and dialogues rooted primarily in sources like Wikipedia, lacking the capability to comprehensively capture various types of knowledge within dialogues. Further comparison of MSDP will be included in our revised paper.
> - Despite both methods involving the extraction of examples from external database to prompt LLMs for knowledge generation and employing In-context learning, our contribution serves as a highly intuitive yet effective extension to MSDP. Besides, our approach also investigates the utilization of generated knowledge. Based on varying levels of knowledge granularity, we propose two distinct approaches for injecting knowledge into dialogues and subsequently compare their respective effects.
>
> **Question 2: No experiment comparing MSDP's performance with in-context learning.**
>
> The MSDP approach has been experimented on the WoW dataset. We will adopt it as a baseline on ODKG dataset. However, since MSDP's backbone model is trained via Megatron[2], which is not open-source, and we want to ensure fairness, so we utilize Flan-T5-XXL, same as KnowEE, to establish a comparison against our proposed framework. We adhere to the experimental settings outlined in the paper and conduct both automated and manual evaluations of MSDP on ODKG. The results are presented as follows:
>
> | Method      | PPL↓      | D-1       | D-2       | Cohe.    | Info.    | Flu.     | Hall.↓   |
> | ----------- | --------- | --------- | --------- | -------- | -------- | -------- | -------- |
> | Flan-T5-XXL | 15.89     | 16.01     | 51.52     | 2.25     | 2.14     | 2.48     | 1.29     |
> | MSDP        | 24.41     | **20.71** | **63.69** | 2.92     | 3.01     | 1.54     | 0.88     |
> | KnowEE-FgKI | **14.25** | 17.88     | **57.33** | **3.09** | 2.83     | 2.43     | 0.96     |
> | KnowEE-CgKI | 15.83     | 17.74     | 56.18     | 3.01     | **3.13** | **2.74** | **0.83** |
>
> From the table, we can observe that MSDP outperforms our framework in terms of diversity (D-1, D-2). However, our approach excels in terms of *coherence* and *informativeness*. Furthermore, MSDP even scores lower in fluency compared to Flan-T5-XXL in zero-shot scenarios. Through case analysis, we find that Flan-T5-XXL with MSDP has a probability of copying content from generated Wikipedia knowledge during text generation, which significantly reduces the fluency of the generated text. In contrast, our method considers various types of knowledge during text generation, which enables us to mitigate this issue more effectively.
>
> **Question 3: No experimental support for the claim that in-context learning actually helps LLMs in inferring emotion, topic, and event knowledge.**
>
> An evaluation of the knowledge generated, such as emotion, topic, and event in the first phase will be conducted to demonstrate the effectiveness of our approach.  We evaluate the accuracy of emotion knowledge generated in the first phase of KnowEE on ED. We select two previous SOTA emotion predict model on this dataset, namely EmpDG[3] and KEMP[4]. The purpose of the experiment is to demonstrate the effectiveness of in-context learning and the similarity-based sampling method. Results are presented as follows.
>
> In the table, "*Prompt Num*" represents the number of examples used for in-context learning (ICL). "*Calculate Similarity*" indicates whether the sampling is done based on the highest sentence similarity score. "*Use NRC-VAD*" indicates whether revised judgment is applied to the outcomes. NRC-VAD[5] is a lexicon comprising VAD vectors with three dimensions (Valence, Arousal, Dominance) for 20,000 English words. Given that numerous different words share similar emotion meanings, such as "happy" and "joyful", if "Use NRC-VAD" is set to "yes," it signifies that we compute the cosine similarity between the predicted emotion and the ground truth emotion's vector in the NRC-VAD dataset. If the computed result exceeds 0.95, the predicted emotion is considered accurate.
>
> | Model  | Prompt Num | Calculate Similarity | Use NRC-VAD | Accuracy（%） |
> | ------ | ---------- | -------------------- | ----------- | ------------- |
> | EmpDG  | --         | --                   | No          | 34.31         |
> | KEMP   | --         | --                   | No          | **39.31**     |
> | KnowEE | 0          | --                   | No          | 3.16          |
> |        | 5          | No                   | No          | 7.76          |
> |        | 8          | No                   | No          | 18.50         |
> |        | 10         | No                   | No          | 19.44         |
> |        | 16         | No                   | No          | 14.53         |
> |        | 5          | Yes                  | No          | 16.96         |
> |        | 8          | Yes                  | No          | 26.14         |
> |        | 10         | Yes                  | No          | 36.29  |
> |        | 16         | Yes                  | No          | 33.17         |
> |        | 5          | Yes                  | Yes         | 75.01         |
> |        | **8**      | Yes                  | Yes         | **84.64**     |
> |        | 10         | Yes                  | Yes         | 79.22         |
> |        | 16         | Yes                  | Yes         | 82.12         |
>
> From the table, it is evident that without utilizing ICL (*Prompt Num = 0*), the model's accuracy is extremely low. However, with the integration of ICL, as shown in rows 5 to 8 of the table, the performance improves to some extent, yet it still falls significantly short of SOTA. By selecting examples with the highest similarity to the current text during the sampling process for ICL, when prompt num is set to 8, the prediction accuracy can approach nearly SOTA (36.29 compared to 39.31). This outcome serves to validate the effectiveness of ICL and the similarity-based sampling method within our approach. Furthermore, following the application of NRC-VAD for revised judgment, the highest accuracy reaches 84.64%. This illustrates that our method ensures the effectiveness of knowledge prediction in practical conversation scenarios.
>
> We are committed to expanding and refining our research based on your suggestions. Your feedback will be carefully considered, with the aim of further enhancing the significance and value of our study.
>
> **References:**
>
> [1] Liu, Zihan, Mostofa Patwary, Ryan Prenger, Shrimai Prabhumoye, Wei Ping, Mohammad Shoeybi, and Bryan Catanzaro. "Multi-stage prompting for knowledgeable dialogue generation." *arXiv preprint arXiv:2203.08745* (2022).
>
> [2] Shoeybi, Mohammad, Mostofa Patwary, Raul Puri, Patrick LeGresley, Jared Casper, and Bryan Catanzaro. "Megatron-lm: Training multi-billion parameter language models using model parallelism." *arXiv preprint arXiv:1909.08053* (2019).
>
> [3] Li, Qintong, Hongshen Chen, Zhaochun Ren, Pengjie Ren, Zhaopeng Tu, and Zhumin Chen. "EmpDG: Multiresolution interactive empathetic dialogue generation." *arXiv preprint arXiv:1911.08698* (2019).
>
> [4] Li, Qintong, Piji Li, Zhaochun Ren, Pengjie Ren, and Zhumin Chen. "Knowledge bridging for empathetic dialogue generation." In *Proceedings of the AAAI Conference on Artificial Intelligence*, vol. 36, no. 10, pp. 10993-11001. 2022.
>
> [5] Mohammad, Saif. "Obtaining reliable human ratings of valence, arousal, and dominance for 20,000 English words." In *Proceedings of the 56th annual meeting of the association for computational linguistics (volume 1: Long papers)*, pp. 174-184. 2018.

---

### Meta-Review · Area_Chair_Lrh3 · 2023-09-19

**Recommendation:** 4

**Metareview:**

Overall this is a solid paper with extensive experiments and convincing results for the proposed framework KnowEE in different knowledge injection settings.

The novelty of he work was somewhat challenged by one of the reviewers, due to missing reference and comparison with MSDP in the original submission. While the authors provided further experiments with MSDP during author response, the reviewer was not fully convinced of positioning of the work as completely novel instead of incremental.

Another criticism around the lack of generalization analysis to real-world scenarios is acknowledged by the authors, but will only be addressed in future work.

---

### Decision · Program_Chairs · 2023-10-07

**Decision:**

Accept-Main

**Comment:**

Overall this is a solid paper with extensive experiments and convincing results for the proposed framework KnowEE in different knowledge injection settings.

The novelty of he work was somewhat challenged by one of the reviewers, due to missing reference and comparison with MSDP in the original submission. While the authors provided further experiments with MSDP during author response, the reviewer was not fully convinced of positioning of the work as completely novel instead of incremental.

Another criticism around the lack of generalization analysis to real-world scenarios is acknowledged by the authors, but will only be addressed in future work.